# Accelerometery-Based Load Symmetry in Track Running Kinematics concerning Body Location, Track Segment, and Distance in Amateur Runners

Antonio Antúnez [1,*], Daniel Rojas-Valverde [2,3,*], Ana Flores-Leonés [1], Carlos D. Gómez-Carmona [1] and Sergio J. Ibáñez [1]

1 Research Group in Optimisation of Training and Sports Performance (GOERD), University of Extremadura, Av. De la Universidad, s/n, 10005 Cáceres, Spain
2 Núcleo de Estudios para el Alto Rendimiento y la Salud (NARS-CIDISAD), Escuela Ciencia del Movimiento humano y calidad de Vida (CIEMHCAVI), Universidad Nacional, Heredia 863000, Costa Rica
3 Clínica de Lesiones Deportivas (Rehab & Readapt), Escuela Ciencia del Movimiento Humano y Calidad de Vida (CIEMHCAVI), Universidad Nacional, Heredia 863000, Costa Rica
* Correspondence: antunez@unex.es (A.A.); drojasv@una.cr (D.R.-V.); Tel.: +506-88250219 (D.R.-V.)

**Abstract:** Background: Previous studies indicate that running at maximum speed on short or curved sections is slower than running on straight sections. This study aimed to analyse the external load symmetry in track running kinematics concerning body location (left vs. right, caudal vs. cephalic), track segment (straight vs. curved) and distance (150 m vs. 300 m). Methods: Twenty experienced athletes ran 150 m and 300 m on an official athletic track and were monitored by Magnetic, Angular Rate and Gravity sensors attached to six different body segments (thorax, lumbar, knees and malleolus). Player Load was quantified as a valid, effective and representative Accelerometery-based variable. Results: (1) Principal component analysis explained 62–93% of the total variance and clustered body locations relevance in curved (knees and malleolus) vs. straight (lumbar, knees, malleolus) running segments; (2) Player Load statistical differences by track segment (curved vs. straight) were found in all body locations; and (3) there were no differences in bilateral symmetries by distance or running segment. Conclusions: Track segment and body location directly impacted accelerometery-based load. Acceleration in straight segments was lower compared to that in curved segments in all the body locations (lumbar, knee and ankle), except in the thorax. Strength and conditioning programs should consider the singularity of curved sprinting (effects of centripetal–centrifugal force) for performance enhancement and injury prevention and focus on the knees and malleolus, as shown in the principal component analysis results.

**Keywords:** accelerometery; inertial measurement units; athletics; speed races; technology

## 1. Introduction

The performance of maximum-speed running races (e.g., 60 m, 100 m, 200 m) has aroused particular research interest. In this type of athletic event, there are very low differences between winning and losing due to its short duration. In this sense, little changes could greatly affect general performance [1,2].

Previous studies have explored several internal and external variables of the athlete that can affect and improve the performance of all-out races. To understand the output in these races, several hypotheses have been proposed. At the physiological level, the maximum running speeds in efforts lasting from a few seconds to several minutes could be accurately predicted from the runner's maximum anaerobic and aerobic speeds [3,4]. On the other hand, other kinematic and kinetic variables influence running economy and technique such as foot–ground reaction, strike patterns, vertical impulse, effective mass, gait length, joint angular trajectories and moments [5]. These variables usually explain how

the athlete modifies their biomechanics to adapt to track conditions, such as curvilinear trajectories (e.g., centrifugal force) or speed [6], and they can be analysed to improve the running technique, adapt injury recovery processes or plan the competition strategy [7].

In addition, between 65 and 80% of runners worldwide suffer injuries due to overload [5,8]. This overload can be due to multiple factors, among which are high rates and magnitudes of loading (e.g., high intensity and volume) [5] or technical deficiencies and structural-functional alterations of the body such as asymmetries (e.g., gait, joint angular trajectories) [9,10]. In this sense, the study of body asymmetries has aroused particular interest in the scientific community due to its importance in sports performance and injury prevention [11,12]. These structural (e.g., joint stability, muscle mass differences) and functional (e.g., joint mobility, muscle stiffness, power, strength) asymmetries can cause changes in the running kinematics and the internal (e.g., functional or physiological outcome of training (heart rate, blood pressure, biochemical release)) and external (e.g., how the load is prescribed or evaluated (acceleration, speed, distance)) loads suffered by the athletes [13,14]. Typically, these asymmetries during running are studied under laboratory and controlled conditions (e.g., treadmill running, ground reaction force platforms, jump power meters) [15–18]. Instead, technology allows for an analysis of these running asymmetries in real conditions [19,20]. Lower limb asymmetries have been studied for their influence on kinematic (e.g., acceleration, deceleration, vectors direction), kinetic (e.g., increase in mediolateral reaction forces) and spatiotemporal (e.g., contact time, flight time, step length and frequency) modifications [21].

The most used method to analyse sprinting performance has been tracking technologies that show the time, distance and speed performed during the race second by second [22]. Thanks to technological advances, data from tracking sensors (e.g., GNSS-GPS) have been improved with the incorporation of accelerometers, gyroscopes and magnetometers that can evaluate external loads non-invasively and segmentally. These sensors allow for the analysis of what happens in each part of the body during sports movements (e.g., running, jumping, changes in speed and direction) [19,20]. Among these devices are the Magnetic, Angular Rate and Gravity (MARGs) sensors, which merge the signals from the sensors to quantify the external load [23–25]. These devices have been used for walking and running analysis in ambulatory settings and real situations in the field. Furthermore, these have been used to quantify the external load on multiple body parts [26,27]. For this purpose, the external load has been quantified by accelerometery-based variables such as the Player Load, which is currently one of the most relevant variables used in the sports area [28].

Running kinematics have been extensively studied in controlled settings (e.g., lab, treadmill), but there is not much evidence when running in other real scenarios (e.g., mountain, track, sand). There is a lack of evidence on how asymmetries during running could be impacted by some contextual variables (e.g., track segment and distance) and body musculoskeletal structures (e.g., body segments or joints). Nowadays, technological advances allow for tracking runners in different body locations. The MARG's sensors are one of those technologies that facilitate runner monitoring during training and competition and require minimum disturbance to the athletes. As a consequence, this study aimed to analyse the external load symmetry in track running kinematics concerning body location (left vs. right, caudal vs. cephalic), track segment (straight vs. curved) and distance (150 m vs. 300 m). We have hypothesised that there are more asymmetries in the external load when running faster and in curvilinear trajectories, and the highest external load is suffered during curved running segments. The outcomes of this study can serve as a basis for improving approaches in terms of the training and sports physical programming of amateur runners. This information could allow technical staff to reorient their personalised plans to the particularities of the running events and the runner's specific kinematic, locomotor and kinetic characteristics.

## 2. Materials and Methods

### 2.1. Study Design

This study was conducted under a cross-sectional and comparative design. The participants ran two distances wearing six Magnetic, Angular Rate and Gravity (MARG) sensors to assess external load during running. The MARG sensors were attached to the runner's body using a special suit to evaluate the changes in external load by body location and assess potential asymmetries between body segments.

Participants ran all-out 150 m and 300 m tests on the official athletics track shown in Figure 1. The total distances were divided into six different segments—every 25 m for the 150 m test and every 50 m for the 300 m test, as shown in Figure 1. The 150 m test starts in a curved 50 m and finishes in a straight 100 m, and the 300 m test starts in a straight 100 m, followed by a curved 100 m, and finishes in a straight 100 m. All participants ran in the first lane.

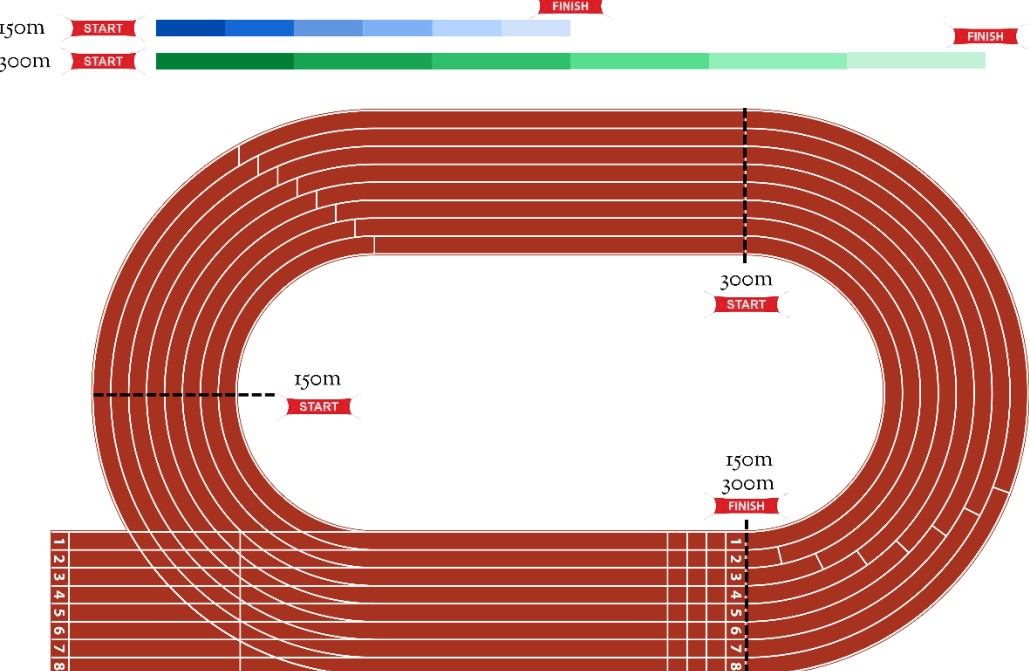

**Figure 1.** Representation of the running segments of the 150 m and 300 m all-out tests.

### 2.2. Participants

A total of 20 male runners (age = 19.9 ± 4.5 years, height = 174.4 ± 6.7 cm, weight = 64.4 ± 9.2 kg, lean mass $_{right leg}$ = 9.3 ± 1.7 kg, lean mass $_{left leg}$ = 9.1 ± 1.6 kg) took part in the study. Participants were track speed runners recruited from the Badajoz Athletic Club (Extremadura, Spain) that met the following inclusion criteria: (1) experience (at least two consecutive competitive years) and (2) training (at least three days/week). The participants competed at the regional and national levels. All participants were described as right dominants.

Participants were allowed to get involved if no neuromuscular, metabolic or structural injuries were reported at least six months before the beginning of the study. Participants were asked to avoid intense training or competitions before the testing. Additionally, the supplementation of stimulants was not allowed two weeks before testing.

Each runner gave their written informed consent and assent according to the Declaration of Helsinki guidelines for biomedical research (18th Medical Assembly 1964, revised in 2013 in Fortaleza). The research protocol was reviewed and approved (reg n° 232/2019) by the institutional review board of the University of Extremadura, Spain. All participants and their parents, if minors, were informed of the details of the study's procedures and the associated risks.

### 2.3. Materials and Procedures

Variables were measured using six MARG sensors (WIMU PRO, Real Track Systems, Almería, Spain). The sensors integrate four three-axis microelectromechanical accelerometers. Two of the accelerometers have the capacity to measure up to $\pm16$ g forces, one has the capacity to measure up to $\pm32$ g forces and one has the capacity to measure up to $\pm400$ g forces. The sensors also included a gyroscope and magnetometers. These sensors have been evaluated in terms of their agreement and reliability in assessing external load in multiple body parts [25,29] in laboratory and field conditions [27]. All calibrations and settings were performed following previously published guidelines [30].

Six MARG sensors were attached, as shown in Figure 2. The MARG's sensors were inserted in the pockets of special spandex suit pants and adjusted using a Velcro system. This attachment system was used to avoid the device's unwanted shaking or vibration [31]. The devices were attached considering previous studies in running protocols [26,32]. One device was located at the second and fourth thoracic vertebrae (~T2–T4). One was located at the first to third lumbar vertebrae (~$L_1$–$L_3$) level. Two devices were attached to bilateral vastus lateralis muscle bellies ($VL_{right}$ and $VL_{left}$), and two devices were fixed to the bilateral 3 cm cephalic from the malleolus peroneus ($MP_{right}$ and $MP_{left}$).

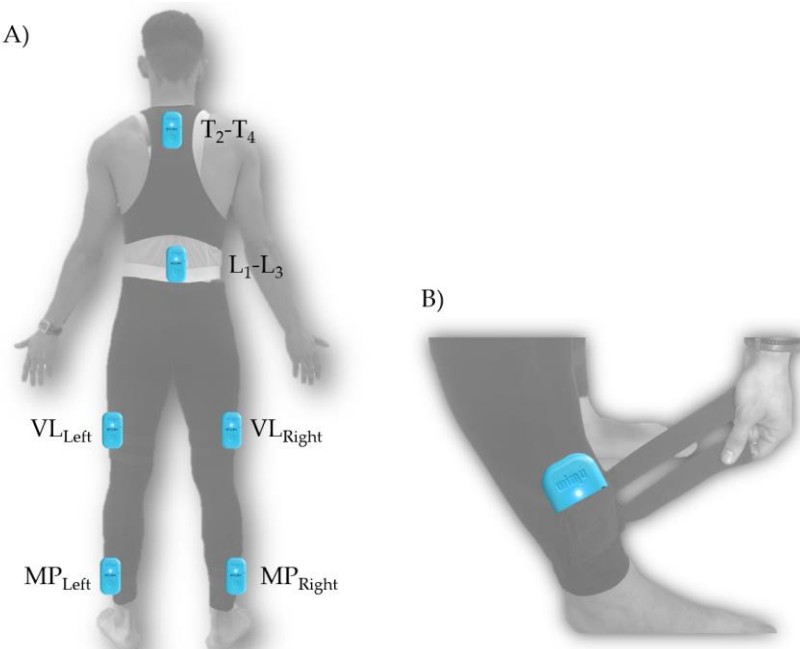

**Figure 2.** Magnetic, angular rate and gravity (MARG) sensors: (**A**) special spandex suit-pants and positioning and (**B**) Velcro straps adjustment system.

Player Load (PL, a.u./min) was chosen as sports' preferred accelerometery-based external workload metric [28,33]. This variable was selected, using objective statistical methods, as the most representative when assessing running external workload [26,32]. This variable is understood as a vector sum of the changes in acceleration in the anterior-posterior (forward), mediolateral (side) and vertical (up) planes. PL is the sum and fusion of the integrated accelerometer's data points in the X, Y and X axes, expressed in arbitrary units [28]. The estimation of PL was performed using the following equation:

$$PL = \sqrt{\frac{(X_n - X_{n-1})^2 + (Y_n - Y_{n-1})^2 + (Z_n - Z_{n-1})^2}{100}}$$

Bilateral asymmetries were considered when differences were lower than 90% [12,14] of the lateral symmetry index, assessed by the following formula [34]:

$$Lateral \ Symetry \ Index = \frac{higher \ value \ (right, left) - lower \ value \ (right, left)}{higher \ value \ (right, left)} * 100$$

Intersegment differences in PL were calculated as the delta's percentage of change (Δ%) between the MP-VL segments. The intersegmental PL was estimated using the following formula:

$$\Delta\% = \frac{(caudal \ segment - cephalic \ segment)}{caudal \ segment} * 100$$

### 2.4. Statistical Analysis

The PL, lateral symmetry index and intersegmental differences were reported using the mean and the standard deviation. The procedure followed before and after the Principal Component Analysis (PCA) was developed according to previous similar papers in sports [35]. The PCA was applied to each track segment of both distances of 150 m (each 25 m, curved vs. straight) and 300 m (each 50 m, curved vs. straight) using the PL of the six MARG sensors. The PL was scaled and centred (Z-Score), PCAs were suitable considering Kaiser–Meyer–Olkin values (KMO = 0.73–0.92) and the Barlett Sphericity test was significant in all PCAs ($p < 0.01$) [36]. After the PCA, eigenvalues greater than one were included for extraction in the respective principal components (PC) [36]. An orthogonal rotation using the VariMax method was used to identify respective loadings in each PC; only loadings greater than 0.6 were retained for interpretation, and the highest loading was reported when a cross-loading was identified between PCs. The PCA outcomes reported were sensors data correlation weights, total variance explained and variance explained by the PC one.

A mixed analysis of variance (MANOVA) was performed to explore the differences between two distances (300 vs. 150) and two track segments (curved vs. straight). Additionally, the symmetries were estimated and compared using a one-way ANOVA by track segment—every 25 m for 150 m and every 50 m for 300 m. A curved vs. straight symmetries comparison was performed using a *t*-test. Considering that the magnitude of the PL could vary between distances, the data were scaled and centred (Z-Score) before this analysis. Alpha was defined in $p < 0.05$. The software Statistical Package for Social Sciences (v.27.0, IBM, Chicago, IL, USA) was used to perform all analyses.

## 3. Results

### 3.1. PCA Analysis of the PL Data of Body Segments by Track Segments

The PCAs of 150 m and 300 m segments explained 62.33–78.15% and 88.22–93.05% of the total variance of the six MARG sensors. The PC one explained 40.72–61.45% in 150 m track segments and 83.07–85.96% in 300 m segments. It has to be highlighted that the curved track segments were represented by $VL_{right}$, $VL_{left}$, $MP_{right}$ and $MP_{left}$, whereas VLright explained the straight track segments, $VL_{left}$, $MP_{right}$, $MP_{left}$ and $L_1$–$L_3$ (see Figure 3).

### 3.2. Comparison of PL Data by Distance, Body Location and Track Segment

The results suggest no differences in PL by distance (relative) in any body segments. Antagonistically, a statistical difference by track segment (curved vs. straight) was found in all body locations, as represented in Table 1.

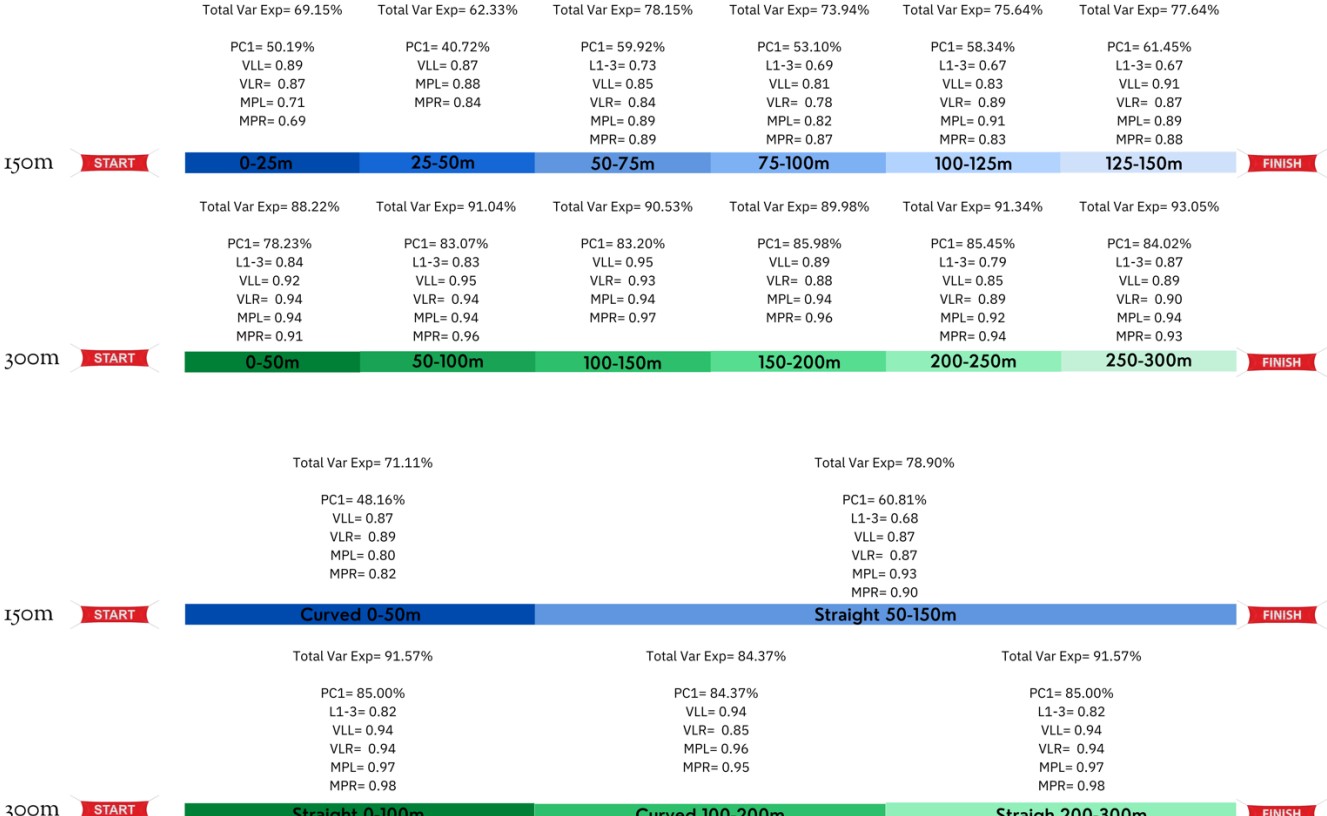

**Figure 3.** Principal components analysis outcomes (orthogonal rotation) based on player load data, with % variance explained by track segment and distance. PC = principal component, L1–3 = lumbar vertebrae 1–3, VLL = vastus lateralis left, VLR = vastus lateralis right, MPL = malleolus peroneus left, MPR = malleolus peroneus right, Var Exp = variance explained.

**Table 1.** Differences in Player Load by body location, distance and track segment.

| Body Location | Distance | Track Segment | | $F_{Track\ segment}$ (*p*-Value) | |
|---|---|---|---|---|---|
| | | Curved | Straight | | |
| $T_2$–$T_4$ | 150 | $0.63 \pm 0.53$ | $0.69 \pm 0.39$ | 0 (1) | |
| | 300 | $0.63 \pm 0.96$ | $0.69 \pm 0.95$ | | |
| $F_{Distance}$ (*p*-value) | | 28.88 (<0.01) | | $F_{Interaction}$ (*p*-value) | 1.25 (0.27) |
| $L_1$–$L_3$ | 150 | $0.75 \pm 0.33$ | $0.68 \pm 0.46$ | 0 (1) | |
| | 300 | $0.75 \pm 0.88$ | $0.68 \pm 0.93$ | | |
| $F_{Distance}$ (*p*-value) | | 40.19 (<0.01) | | $F_{Interaction}$ (*p*-value) | 1.24 (0.27) |
| $VL_{right}$ | 150 | $0.79 \pm 0.30$ | $0.71 \pm 0.39$ | 0 (1) | |
| | 300 | $0.79 \pm 0.80$ | $0.71 \pm 0.91$ | | |
| $F_{Distance}$ (*p*-value) | | 50.00 (<0.01) | | $F_{Interaction}$ (*p*-value) | 2.51 (0.12) |
| $VL_{left}$ | 150 | $0.76 \pm 0.30$ | $0.73 \pm 0.35$ | 0 (1) | |
| | 300 | $0.76 \pm 0.86$ | $0.73 \pm 0.91$ | | |
| $F_{Distance}$ (*p*-value) | | 46.23 (<0.01) | | $F_{Interaction}$ (*p*-value) | 1.43 (0.24) |
| $MP_{right}$ | 150 | $0.75 \pm 0.36$ | $0.72 \pm 0.42$ | 0 (1) | |
| | 300 | $0.75 \pm 0.85$ | $0.72 \pm 0.88$ | | |
| $F_{Distance}$ (*p*-value) | | 45.32 (<0.01) | | $F_{Interaction}$ (*p*-value) | 0.43 (0.52) |
| $MP_{left}$ | 150 | $0.74 \pm 0.39$ | $0.71 \pm 0.38$ | 0 (1) | |
| | 300 | $0.74 \pm 0.86$ | $0.71 \pm 0.92$ | | |
| $F_{Distance}$ (*p*-value) | | 41.84 (<0.01) | | $F_{Interaction}$ (*p*-value) | 0.57 (0.46) |

*3.3. Symmetries and Intersegmental Differences of PL Data by Distance, Body Location and Track Segment*

The results indicated no difference in the bilateral or intersegmental symmetries by distance (see Table 2). Additionally, there were no differences in bilateral symmetry between the curved and straight segments in 150 m ($10.16 \pm 8.08$ vs. $10.00 \pm 6.15$; $t = 0.09$, $p = 0.93$) or 300 m ($7.31 \pm 6.86$ vs. $7.91 \pm 5.05$; $t = 0.38$, $p = 0.71$). The intersegmental difference presented no differences in 150 m ($24.43 \pm 11.17$ vs. $25.59 \pm 8.21$; $t = 0.63$, $p = 0.63$) or 300 m ($20.65 \pm 8.21$ vs. $18.90 \pm 8.58$; $t = 1.25$, $p = 0.23$).

**Table 2.** Symmetries percentage comparison of Player Load by distance segments.

| Symmetry | Distance | | | | | | $F_{Interaction}$ (*p*-Value) |
|---|---|---|---|---|---|---|---|
| 150 m speed race | 0–25 m | 25–50 m | 50–75 m | 75–100 m | 100–125 m | 125–150 m | |
| Bilateral symmetry | | | | | | | |
| Malleolus Peroneus | $11.92 \pm 12.46$ | $11.46 \pm 6.99$ | $10.74 \pm 8.49$ | $13.05 \pm 7.93$ | $11.33 \pm 7.83$ | $12.74 \pm 8.42$ | 0.21 (0.96) |
| Vastus Lateralis | $11.28 \pm 7.11$ | $13.87 \pm 20.96$ | $8.91 \pm 8.64$ | $11.82 \pm 6.68$ | $11.24 \pm 8.11$ | $8.92 \pm 5.79$ | 0.62 (0.68) |
| Intersegmental difference | | | | | | | |
| MP–VL | $25.51 \pm 10.89$ | $26.11 \pm 9.94$ | $25.37 \pm 10.89$ | $25.34 \pm 11.13$ | $27.17 \pm 9.09$ | $23.61 \pm 8.50$ | 0.50 (0.78) |
| 300 m speed race | 0–50 m | 50–100 m | 100–150 m | 150–200 m | 200–250 m | 250–300 m | |
| Bilateral symmetry | | | | | | | |
| Malleolus Peroneus | $8.45 \pm 8.57$ | $7.81 \pm 5.86$ | $7.67 \pm 6.57$ | $8.41 \pm 6.91$ | $11.19 \pm 6.91$ | $8.34 \pm 4.14$ | 0.86 (0.52) |
| Vastus Lateralis | $8.21 \pm 6.35$ | $6.11 \pm 5.29$ | $7.47 \pm 5.69$ | $8.87 \pm 7.36$ | $9.64 \pm 5.32$ | $7.68 \pm 6.91$ | 0.98 (0.44) |
| Intersegmental difference | | | | | | | |
| MP–VL | $20.81 \pm 10.53$ | $20.03 \pm 9.40$ | $20.00 \pm 9.57$ | $20.56 \pm 8.79$ | $18.13 \pm 8.51$ | $18.08 \pm 8.94$ | 0.74 (0.60) |

## 4. Discussion

This study aimed to analyse the acceleration symmetry by body location (left vs. right, caudal vs. cephalic), track segment (straight vs. curved) and distance (150 m vs. 300 m) during track running. We have hypothesised that (1) there are more asymmetries in the external load when running faster and in curved trajectories and (2) the highest external load is suffered during curved running segments.

The results of the present study indicate differences in the magnitude of the variables based on accelerometery (PL) during the race in curved segments compared to straight segments. These differences are evident bilaterally in body locations such as the lumbar, knees and ankles. Additionally, there are no differences in bilateral or intersegmental symmetry related to distance or track segments. Therefore, the first two hypotheses are rejected, and the third is accepted.

In this sense, previous studies indicate that running at maximum speed on short and curved sections is slower than running on straight sections. This is produced by two factors that decrease such as step frequency and turning radius [37]. Additionally, this speed reduction during curved running has been attributed to kinematic (e.g., acceleration, deceleration, vectors direction), kinetic (e.g., increase in mediolateral reaction forces) and spatiotemporal (e.g., contact time, flight time, step length and frequency) modifications [21]. In this sense, the speed reduction in curves is caused by an increase in lateral ground reaction forces leading to a decrease in peak vertical ground reaction [38].

In this study, speed loss differed significantly between runners in curved segments. This fact indicates, from a biomechanical point of view, that there are good curved runners and bad curved runners [39]. It is suggested that there are runners that are able to accommodate a tighter radius than others [37]. Therefore, even though it is inevitable that the biomechanics and kinematics of running undergo changes in curved segments, as identified in this study (usually due to the increased load demand), there are proposals to improve the technique and minimise these differences. It should be considered that this outcome should be applied to amateur runners given the spatiotemporal and musculoskeletal differences with elite runners [40].

During curved running, the medial-lateral reaction ground forces and impulse increase compared to straight running. This could result from counteracting the suffered centrifugal force during curved running [39]. This phenomenon was evident in this study's outcomes, which suggest that the accelerometery load was higher during curved running, identified by the ankles, knees and lumbar body locations.

Evidence suggests that there could be symmetric kinematic modifications such as a greater peak hip adduction, peak hip internal rotation and peak ankle eversion on the curved segment compared to the straight segment [21]. The results of the present study may suggest that the counteract of the centrifugal forces performed during curved running could result in a similar external load, as found in this study. This statement is supported by evidence suggesting that, despite asymmetries during running affecting kinematics, these changes are too small to affect the running economy [41]. Additionally, some plastic compensations in the musculoskeletal system could mitigate this potential influence of asymmetry [41]. The decrease in running speed during the curved running may be necessary to avoid a higher load or a difference between the load segments and thus avoid injury or reduce the large gravitational forces caused by the centrifugal force in these running segments. In this sense, evidence supports the idea that running speed does not influence the lower limb's kinematics asymmetry among amateurs [42]. This is why it is recommended that strength and conditioning programming aim to work on the hips, ankles and feet in the non-sagittal planes (e.g., proprioception, power, strength and positioning) [21,43].

The outcomes of this study indicate that those body compensations could preserve accelerometery load stability during curved running, although curved running is considered asymmetrical in nature [44]. It is recommended that, to maintain good performance during curved paths during the race, it is critical to perform at a similar speed, flight distance and stride length compared to straight paths [39]. It is necessary to achieve an optimal technique in curved segments, maintaining similar kinematics and kinetics in the sagittal plane as on a straight running [37,39].

Additionally, one of the highlights of this study was the use of PCA to cluster the MARG sensors based on their critical role in both curved and straight running. The results suggested that PCA explained 62–93% of the total variance and clustered body locations relevance in curved (knees and malleolus) vs. straight (lumbar, knees, malleolus) running segments. This outcome implies that the lumbar region may have a critical role during straight running compared to curved sections; it is particularly sensitive to the body centre of mass movement [41]. This may be due to the greatest vertical reaction forces provoked during straight running compared to the mediolateral reaction forces caused during curved running. In this sense, studies have shown that mean peak vertical and resultant forces usually decrease during curved running. The left step has the greatest decrease compared to the right [44]. Additionally, the upper limbs' static and dynamic asymmetries during curved vs. straight running required a more in-depth analysis due to the potential influence of accelerometery load during running [41].

### 4.1. Limitations

While this study presents some evidence of the external load differences and symmetries during running by distance and track segments, these findings must be interpreted considering some limitations. Due to the sample characteristics (e.g., level of runners), the outcomes of this study cannot be applied to other populations (e.g., professional athletes) because there are technical differences that could modify the results obtained.

It is important to note that, due to logistical reasons, it was not possible to experiment with official World Athletics competition distances (e.g., 400 m, 200 m). Additionally, we selected the first lane of the track for all assessments, but the results could vary depending on the lane used. More analysis should be conducted exploring the differences in accelerometery and external load depending on the centripetal forces suffered based on the lane used (ratio of the track curve). Finally, the sample size and potential heterogeneity can cause

relatively high standard deviations in the results, which should be improved in future studies. Additionally, it should be explored more in depth if the asymmetry indices used in gait analysis and commonly applied to running movement best characterise running asymmetry [9].

### 4.2. Practical Applications

The outcomes of this study suggest that the strategies of speed races could vary according to the distance and the track segment (curve vs. straight). Considering that the sprint testing ability typically focuses on linear speed, evaluation should be made using curved running. Additionally, curved segments must be trained from an external load point of view to withstand loads during high-speed actions such as 150 m and 300 m races. During training, the optimal biomechanical sprinting angle, body position, step length and flight distance should be assessed to suffer less external load during running in curved trajectories [39]. Additionally, the athletics track running lane should be considered when planning a running strategy due to the potential differences in speed and centrifugal forces [37,45,46].

The MARGs sensors have proven to be an effective technology for tracking multi-segmented external load during running in-field testing. Additionally, this method of evaluation allows for the measurement of different body locations simultaneously, employing accelerometery with a non-invasive method, and in real situations of competition or training.

## 5. Conclusions

The results suggest differences in the accelerometery-based load in running (Player Load) when comparing the curved and straight segments during running on an athletic track. It seems that the straight segments present less of an accelerometery-based load than the curved segments in all the body locations (lumbar, knee and ankle), except for the thorax. These results confirmed that centrifugal force during curved segments provokes the athlete to perform a mediolateral ground-reaction force to counteract this force and stay balanced during running. Additionally, there are no differences in bilateral symmetry between the magnitude of the devices placed in the malleolus peroneus or vastus lateralis at any distance (150 m or 300 m).

Principal component analysis suggests that the lumbar region may have a critical role during straight running compared to curved sections. This may be due to the greatest vertical reaction forces provoked during straight running compared to the mediolateral reaction forces caused during curved running.

New technology should be developed to evaluate the ground reaction forces, running angles and speed in a simple, efficient and effective manner. Future research may focus on the effect of training strategies to reduce or counteract the centrifugal force. Additionally, these future studies may use available technology to give real-time feedback to the runner considering the external load suffered during curved running.

**Author Contributions:** Conceptualisation, A.A., A.F.-L. and C.D.G.-C.; methodology, A.A., A.F.-L. and C.D.G.-C.; software, C.D.G.-C.; validation, D.R.-V. and C.D.G.-C.; formal analysis, D.R.-V. and C.D.G.-C.; investigation, A.A., A.F.-L. and C.G-C; resources, A.A. and S.J.I.; data curation, D.R.-V. and C.D.G.-C.; writing—original draft preparation, D.R.-V. and A.F.-L.; writing—review and editing, A.A., D.R.-V., A.F.-L., C.D.G.-C and S.J.I.; visualisation, D.R.-V. and C.D.G.-C.; supervision, A.A., D.R.-V. and S.J.I.; project administration, A.A. and S.J.I.; funding acquisition, A.A., D.R.-V. and S.J.I. All authors have read and agreed to the published version of the manuscript.

**Funding:** This work was led by the Group for the Optimisation of Training and Sports Performance (GOERD) of the Faculty of Sports Sciences of the University of Extremadura and has been partially supported by the funding for research groups (GR21149) granted by the Government of Extremadura (Employment and Infrastructure office—Consejería de Empleo e Infraestructuras), with the contribution of the European Union through the European Regional Development Fund (ERDF). The article

resulted from a project economically supported by the National University of Costa Rica Research vice rectory.

**Informed Consent Statement:** Informed consent was obtained from all subjects involved in the study.

**Data Availability Statement:** The data bases are available if required by contacting the corresponding author.

**Acknowledgments:** The authors are grateful for the Assistance to Research Groups (GR21149) of the Regional Government of Extremadura (Ministry of Employment and Infrastructure), with the contribution of the European Union through the European Regional Development Funds (ERDF).

**Conflicts of Interest:** The authors declare no conflict of interest.

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
