# Peer review of "Accelerometery-Based Load Symmetry in Track Running Kinematics concerning Body Location, Track Segment, and Distance in Amateur Runners"

_symmetry, doi:10.3390/sym14112332_

Round 1

Reviewer 1 Report

General comments

This manuscript aims at analyzing the external load symmetry in track running kinematics concerning body location (left vs. right, caudal vs. cephalic), track segment (straight vs. curved) and distance (150 m vs. 300m). Research is relevant and well-performed. Yet, some specific and minor comments detailed below should be addressed to make MS worth of acceptance for publication,

Specific comments

Relevant missing ref:

https://pubmed.ncbi.nlm.nih.gov/24086316/

Was athletics track’s first lane used?

Why isn’t PL measurement unit m/s^2 (viz., acceleration m. u.)?

Do you mean lumbar sensor is particularly sensitive to body centre of mass movement? If yes, please, state it explicitly.

Upper limbs static and/or dynamic (read above ref, as well) asymmetries could matter during curved running. Please, comment.

Minor comments

(lines 25÷7) Please, re-phrase;

(l45÷9) please, split;

(l148 and elsewhere throughout MS) please, do not start sentences with acronyms;

Reff 25 and 35 are the same;

(l185 and elsewhere throughout MS) please, do not use acronyms in headings;

(l189) … and MPleft, whereas VLright (<– check subscript) explained…

(l208) please, check font size;

(l227-8) I understand what you mean, but doubt that English language is correct. Please, re-phrase.

Author Response

Dear Editor and reviewers:

We have carefully considered all reviewers' recommendations for the paper (Manuscript ID: symmetry-1880783) entitled "Accelerometry-based load symmetry in track running kinematics in relation to body location, track segment, and distance”. Please find enclosed our detailed answers to reviewers' queries. The authors declare that the manuscript is original and has not been considered for publication elsewhere. Additionally, the authors had approved the paper for release and agree with its content.

Please find all corrections in color inside the manuscript.

Reviewer 1.

General comments

This manuscript aims at analyzing the external load symmetry in track running kinematics concerning body location (left vs. right, caudal vs. cephalic), track segment (straight vs. curved) and distance (150 m vs. 300m). Research is relevant and well-performed. Yet, some specific and minor comments detailed below should be addressed to make MS worth of acceptance for publication,

R/ As Corresponding Author On Behalf Of All Authors We Thank The Reviewers For Their Contributions To Improve The Quality Of This MS.

Specific comments

Relevant missing ref:

https://pubmed.ncbi.nlm.nih.gov/24086316/

R/ We really agree with the reviewer and we have included this reference.

Was athletics track’s first lane used?

R/ We want to thank the reviewer for his/her recommendation to included this information.

Why isn’t PL measurement unit m/s^2 (viz., acceleration m. u.)?

R/ Thank you for your observation. This variable the result of the vector sum of the changes in acceleration in the anterior–posterior (forward) medio-lateral (side) and vertical (up) planes. This is why it is not expressed in m/s2.

Do you mean lumbar sensor is particularly sensitive to body centre of mass movement? If yes, please, state it explicitly.

R/ Thank you for your recommendation, it was corrected.

Upper limb’s static and/or dynamic (read above ref, as well) asymmetries could matter during curved running. Please, comment.

R/ Thank you for your recommendation, it was corrected.

Minor comments

(lines 25÷7) Please, re-phrase;

R/ This sentences was corrected considering this suggestion

(l45÷9) please, split;

R/It was corrected considering this suggestion

(l148 and elsewhere throughout MS) please, do not start sentences with acronyms;

R/ Thank you for highlighting this issue, we have corrected it.

Reff 25 and 35 are the same;

R/ Thank you for highlighting this issue, we have corrected it.

(l185 and elsewhere throughout MS) please, do not use acronyms in headings;

R/ Thank you for highlighting this issue, we have corrected it.

(l189) … and MPleft, whereas VLright (<– check subscript) explained…

R/ Thank you for highlighting this issue, we have corrected it.

(l208) please, check font size;

R/ Thank you for highlighting this issue, we have corrected it.

(l227-8) I understand what you mean, but doubt that English language is correct. Please, re-phrase.

R/ Thank you for highlighting this issue, we have corrected it.

Reviewer 2

This manuscript entitled “Accelerometry-based load symmetry in track running kinematics in relation to body location, track segment, and distance” primarily aimed to analyze the external load symmetry in track running kinematics concerning body location (left vs right, caudal vs cephalic), track segment (straight vs curved) and distance (150m vs 300m). The authors bring an interesting study, but there are still some problems that cannot up this study to a publishing level. Some suggestions are listed in the specific comments below.

R/ As Corresponding Author On Behalf Of All Authors We Thank The Reviewers For Their Contributions To Improve The Quality Of This MS.

 Specific comments:

What do internal and external loads mean respectively? What is the relationship between them? Please explain.

R/ We thank the reviewer for the opportunity to expand and warm up. We consider that adding a definition and examples can make the concept of internal and external load clearer.

Why is it necessary to analyze the influence of factors such as track segment in this study? In addition, the authors should further emphasize the necessary and potential value of this study.

R/ we have completed and added reference information to the subject of study, in addition, the explanation on the relevance/contribution of the study in the area has been extended.

  1. Did the authors investigate the dominant leg in all subjects? Were they all consistent? This factor may affect the results of this study due to the curved track segment and symmetry.

R/ We really appreciate the reviewer for this comment. All participants were right-hemisphere dominants, it was clarified in the text.

  1. ‘Also, this speed reduction during curved running has been attributed to kinematic, kinetic and spatiotemporal modifications’, how do kinematic, kinetic and spatiotemporal modifications lead to a reduction in speed during curved running?

R/ We thank the reviewer for the opportunity to clarify this point. Arguments and qualifications have been added to amplify this statement.

  1. ‘The results of the present study may suggest that the counteract of the centrifugal forces performed during curved running could result in a similar external load as found in this study.’, more evidence needs to be provided to support the authors’ conclusions.

R/ We really appreciate the reviewer request, we have included some evidence that support this statement.

  1. ‘Due to the sample characteristics (e.g., level of runners), the outcomes of this study cannot be applied to other populations…’, as stated by the authors the sample of this study was strictly characterized and it is recommended that the authors emphasize this in the title.

R/We really appreciate the reviewer’s recommendation. We have consider it and we included in the title that the sample analyzed were amateur runners as stated in the method section.

  1. In summary, please ensure that your manuscript is prepared correctly (without any grammatical and spelling mistakes) and formatted before submitting a revision.

R/ The manuscript was corrected in some grammar and spelling mistakes and formatted based on journal guidelines. We really appreciate the reviewer’s consideration of the MS.

Reviewer 2 Report

Review comment

This manuscript entitled “Accelerometry-based load symmetry in track running kinematics in relation to body location, track segment, and distance” primarily aimed to analyze the external load symmetry in track running kinematics concerning body location (left vs right, caudal vs cephalic), track segment (straight vs curved) and distance (150m vs 300m). The authors bring an interesting study, but there are still some problems that cannot up this study to a publishing level. Some suggestions are listed in the specific comments below.

Specific comments:

1.     What do internal and external loads mean respectively? What is the relationship between them? Please explain.

2.     Considering that the purpose of this study involves the effect of track segment (straight vs curved) on the symmetry of running kinematics, the current introduction section lacks the relevant research background. Why is it necessary to analyze the influence of factors such as track segment in this study? In addition, the authors should further emphasize the necessary and potential value of this study.

3.     Did the authors investigate the dominant leg in all subjects? Were they all consistent? This factor may affect the results of this study due to the curved track segment and symmetry.

4.     ‘Also, this speed reduction during curved running has been attributed to kinematic, kinetic and spatiotemporal modifications’, how do kinematic, kinetic and spatiotemporal modifications lead to a reduction in speed during curved running?

5.     ‘The results of the present study may suggest that the counteract of the centrifugal forces performed during curved running could result in a similar external load as found in this study.’, more evidence needs to be provided to support the authors’ conclusions.

6.     ‘Due to the sample characteristics (e.g., level of runners), the outcomes of this study cannot be applied to other populations…’, as stated by the authors the sample of this study was strictly characterized and it is recommended that the authors emphasize this in the title.

7.     In summary, please ensure that your manuscript is prepared correctly (without any grammatical and spelling mistakes) and formatted before submitting a revision.

Author Response

(The authors gave the same response as above.)

Round 2

Reviewer 1 Report

General comments

Authors addressed all my issues sufficiently enough.

Specific comment

Original comment: “Was athletics track’s first lane used?”

Maybe, third or fourth are more representative.

Overall, authors did not address this issue.

Author Response

Dear Editor and reviewers:

We have carefully considered all reviewers' recommendations for the paper (Manuscript ID: symmetry-1880783) entitled "Accelerometery-based load symmetry in track running kinematics in relation to body location, track segment, and distance”. Please find enclosed our detailed answers to reviewers' queries. The authors declare that the manuscript is original and has not been considered for publication elsewhere. Additionally, the authors had approved the paper for release and agree with its content.

Please find all corrections in color inside the manuscript.

Reviewer

General comments

Authors addressed all my issues sufficiently enough.

 R/Dear editors and reviewers, we really appreciate your review and comments to improve the final version of the MS. It really give a boost in the final quality of the document.

Specific comment

Original comment: “Was athletics track’s first lane used?” Maybe, third or fourth are more representative.

R/Based on the first review of the MS, we clarified the following in methods section: ¨ All participants ran in the first lane.¨

Overall, authors did not address this issue.

R/We understand the concern and significant contribution of the reviewer. In this case, it is a methodological aspect that we do not consider due to logistical aspects. We have included this potential limitation in the appropriate study limitations section.